# The effect of COVID-19 on poor treatment control among ambulatory Hypertensive and/ or Diabetic patients in Northwest Ethiopia

**Tadesse Awoke Ayele[1], Habtewold Shibru[2], Malede Mequanent Sisay[1], Tesfahun Melese[3], Melkitu Fentie[4], Telake Azale[5], Tariku Belachew[6], Kegnie Shitu[5], Tesfa Sewunet Alamneh[1]***

**1** Epidemiology & Biostatistics Department, College of Medicine and Health Sciences, University of Gondar, Gondar, Ethiopia, **2** Internal Medicine Department, College of Medicine and Health Sciences, University of Gondar, Gondar, Ethiopia, **3** Health Informatics Department, College of Medicine and Health Sciences, University of Gondar, Gondar, Ethiopia, **4** Nutrition Department, College of Medicine and Health Sciences, University of Gondar, Gondar, Ethiopia, **5** Health Education & Behavioural Sciences Department, College of Medicine and Health Sciences, University of Gondar, Gondar, Ethiopia, **6** Amhara Health Bureau, Bahir-Dar, Ethiopia

* tesfasewunet23@gmail.com

**Data Availability Statement:** All relevant data are within the paper and its Supporting Information files.

## Abstract

### Background

Diabetes and hypertension have emerged as important clinical and public health problems in Ethiopia. The need to have long-term sustainable healthcare services for patients with diabetes and hypertension is essential to enhance good treatment control among those patients and subsequently delay or prevent complications. A collective shift towards acute care for COVID-19 patients combined with different measures to contain the pandemic had disrupted ambulatory care. Hence, it is expected to have a significant impact on treatment control of hypertensive and diabetic patients. However, there is limited evidence on the effect of the pandemic on treatment control and its determinants. Therefore, this study aimed to assess the effect of COVID-19 pandemic on treatment control of ambulatory Hypertensive and Diabetic patients and identify the factors for poor treatment control in North West Ethiopia.

### Methods

A retrospective chart review and cross-sectional survey design were conducted between December 2020 and February 2021. Using a stratified systematic random sampling technique, 836 diabetic and/or hypertensive patients were included in the study. Web-based data collection was done using Kobo collect. The changes in the proportion of poor treatment control among ambulatory Hypertensive and/or Diabetic patients during the COVID-19 pandemic period were assessed. A multivariable binary logistic regression mixed model was fitted to identify the determinants of poor treatment control. The odds ratios were reported in both crude and adjusted form, together with their 95% confidence intervals and p-values.

**Funding:** This study was funded by Ethiopian Ministry of Health with grant number of 34/49/1142 where professor Tadesse Awoke Ayele was the grant recipient. however, the funders had no role in study design, data collection and analysis, decision to publish, or preparation of the manuscript.

**Competing interests:** The authors have declared that no competing interests exist.

**Abbreviations:** AOR, Adjusted Odds Ratio; BID, two times per day; IQR, Inter Quartile Range; LLR, Likelihood Ratio; TID, three times per day; QID, four times per day.

## Result

Poor treatment control increased significantly from 24.81% (21.95, 27.92) prior to the COVID-19 pandemic to 30.33% (27.01, 33.88), 35.66% (32.26, 39.20), 36.69% (33.40, 40.12), and 34.18% (3102, 37.49) in the first, second, third, and fourth months following the date of the first COVID-19 case detection in Ethiopia, respectively. Marital status (AOR = 0.56, 95%CI; 0.41, 0.74), regimen of medication administration (AOR = 1.30, 95%CI; 1.02, 166), daily (AOR = 0.12, 95%CI; 0.08, 0.20), twice (AOR = 0.42, 95%CI; 0.30. 0.59), and three times (AOR = 0.31, 95%CI; 0.21, 0.47) frequency of medication, number medications taken per day (AOR = 0.79, 95%CI;0.73, 0.87), patients habits like hazardous alcohol use (AOR = 1.29, 95%CI; 1.02, 1.65) and sedentary lifestyle (AOR = 1.72,95%CI;1.46, 2.02), missed appointment during the COVID-19 pandemic (AOR = 2.09, 95%CI; 1.79, 2.45), and presence of disease related complication (AOR = 1.11, 95%CI; 0.93, 1.34) were significantly associated with poor treatment control among Diabetic and/or hypertensive patients during the COVID-19 pandemic.

## Conclusion

The COVID-19 pandemic had a substantial impact on ambulatory Diabetic and/or Hypertensive patients' treatment control. Being married, as well as the frequency and types of medicines taken per day were all found to be negatively associated with poor treatment control. During the COVID -19 pandemic, patients' habits such as hazardous alcohol use and sedentary lifestyle, longer follow-up time, having disease-related complication (s), patients taking injectable medication, number of medications per day, and missed appointments were positively associated with poor treatment control in ambulatory diabetic and hypertensive patients. Therefore, it is better to consider the risk factors of poor treatment control while designing and implementing policies and strategies for chronic disease control.

## Background

The highly contagious SARS-CoV-2 virus has killed over 4 million people globally since its detection in Wuhan, China, in December 2019 [1]. The COVID-19 epidemic has far-reaching consequences for the healthcare system. The COVID-19 pandemic has a substantial impact on ambulatory follow-up care in Africa's, particularly Ethiopia's, underdeveloped healthcare system. Governments from many nations have been obliged to enact legislative measures to stem the spread of the infection, which include complete lockdown, social isolation, confinement at home, and the suspension of all business activities varying in duration and extent [2, 3].

In Ethiopia, diabetes and hypertension have emerged as major clinical and public health challenges. Both are significant causes of premature death and morbidity. To avoid poor glycaemic or blood pressure control, and hence delay or prevent disease progression and accompanying consequences, individuals with diabetes and hypertension require long-term sustainable healthcare services. Clinical services have been hampered by the necessity for contact precautions to stop the spread of the COVID-19 pandemic [3, 4].

People with chronic medical conditions, such as diabetes and/or hypertension, are particularly prone to infection and have a higher rate of morbidity and mortality as a result of infection. COVID-19 infection was two to three times more likely in people with diabetes or

hypertension, and it was associated with worse outcomes and a higher mortality rate. Diabetic individuals, for example, have twice the chance of being admitted to the intensive care unit for COVID-19 infection as non-diabetic patients. Diabetic and hypertensive patients are more susceptible to COVID-19 infection due to a higher risk of infection as a result of leukocyte malfunction, a pro-inflammatory profile, and micro-angiopathic alterations affecting the lungs or Angiotensin converting enzyme -2 receptors [5, 6].

The COVID-19 pandemic is expected to have a substantial impact on hypertensive and diabetic patients' treatment control. Because it allows for the cancellation of non-emergent treatments and clinical appointments regularly, which will have an impact on the clinical outcomes for patients with chronic medical illnesses [7–9]. Meanwhile, appropriate treatment controls assist delay or preventing problems, as well as reducing COVID-19 infection morbidity and mortality. As a result of the pandemic's impact, some centres have made rapid and urgent shifts toward alternative patient care methodologies, such as virtual encounters (via video or phone) and medication delivery via mail, which are thought to reduce the risk of infection transmission and the burden on the health-care system. The number of ambulatory visits has decreased by 30% in some locations compared to pre-COVID-19 period [2, 5, 10, 11]

Patients' adherence to healthy lifestyles and medications has been influenced by various measures attempted to manage the pandemic, such as lockdowns. Age, sex, non-adherence to drugs, non-adherence to dietary restrictions, physical inactivity, the number of medications taken, and the existence of co-morbid disease are all known to play a role in diabetes and hypertension control. Factors associated with the COVID-19-related economic and social crisis may have a substantial indirect impact on the care [5, 11, 12].The pandemic had a huge impact, especially on these patients, where the goal isn't to cure them completely, but rather to slow down the disease's progression and prevent complications. However, there is no evidence of the pandemic's impact on medication adherence and its determinants in diabetic and hypertensive patients. Therefore, this study aimed at assessing treatment control during COVID-19 pandemic periods and its factors. Thus, assessing the treatment control of patients with diabetes and/ or hypertension during the pandemic might have paramount importance for designing and implementing intervention measures.

## Methods

### Study design and setting

Retrospective chart review and facility-based cross-sectional study design were employed in public Hospitals that are giving chronic care in Northwest Ethiopia, from January to March 2021. Northwest Ethiopia includes Amhara regional state which has 15 Zones and 180 whereas (139 rural and 41 urban). The first COVID-19 case was confirmed on March 30, 2020. Treatment centres, isolation, and quarantine centres were established in the region as the COVID 19 prevention and treatment strategies. According to the regional health COVID 19 command team report currently, 291,148 susceptible individuals were tested for COVID 19. Of these 11727 cases were detected, 2862 recovered, and 293 death recorded in the region. The region has 80 hospitals (6 referrals, 2 generals, and 72 primaries), 847 health centres, and 3,342 health posts. The study was incorporating all hospitals (referral, district) in the Amhara region with chronic care centres.

### Source and study population

The source population consisted of all patients with diabetes and/or hypertension who had follow-ups at hospitals in the Amhara regional state. The study population consisted of patients who had chronic care appointments and follow-up during the data collection period. Patients

with common chronic disease conditions who were at least 18 years old and had been on medication for at least 2 years were included in the study. Participants who returned within the data collection period were not included in the study.

## Sample size and sampling procedures

With the key objectives in mind, the sample size was estimated using the single population proportion formula. A design effect of 2 and a 10% non-response rate were also taken into account. As a result, the ultimate sample size was estimated to be 845 patients.

To recruit study participants, all referral and selected district hospitals in the region were included in the sampling procedure. First, stratification was done based on the status of the hospital (referral or district). Hospitals were then chosen from each stratum. Finally, study participants were chosen using a systematic random sampling technique in the specified hospitals based on disease type. The total sample was proportionally allocated across illness types.

## Data collection methods and measurements

To collect the required data on the variables of interest, primary and secondary sources were used. The baseline measurement was taken from the most recent measurement before the first date of COVID-19 case detection, and one year following the COVID-19 case detection was divided into four periods, with the first three months, second three months, third three months, and fourth three months being the first, second, third, and fourth periods, respectively.

Charts were retrieved from the treatment centres in the selected Hospitals. During data collection, it has been about a year since the emergence of the COVID-19 pandemic. One-year retrospective data was extracted for the same patients before the COVID-19 pandemic. All available epidemiological information was collected including, socio-demographic variables, clinical factors, and patient treatment control. Treatment control: was ascertained by the treating physician working in the respective follow-up clinics as poor or good. Poor treatment control was considered when the treatment target was not achieved on that specific follow-up date. The treating physician used glycaemic target or blood pressure target coupled with other clinical parameters to ascertain treatment control. Missed appointments: when a patient did not attend the follow-up according to the physicians' schedule.

Health management information systems and patient charts were used to extract the data using chart extraction form. Patients' interview was made after the appointment logbook and patient chart retrieval.

## Quality assurance mechanism

Data collectors and supervisors were provided training to maintain the data's quality. Medical physicians and other trained health professionals who work in treatment centres were recruited. The questionnaire was translated into Amharic, the native language, and then returned to English to ensure consistency. After the questionnaire was converted to electronic data from using Kobo-collect, web-based data collection was done. A pre-test was conducted, and possible adjustments were made, as well as an internal consistency reliability test. The collected data were checked for completeness and consistency daily at the server.

## Data management and analysis

Following completion of data collection, the web-based data was exported to STATA and R for management and analysis. Cleaning, coding, categorization, and error inception were made by

the research team. Results were explored using descriptive statistical techniques and prevalence, mean, median, inter-quartile range, and standard deviations were computed.

Since the data had hierarchical nature, it could violate the independence of observations and equal variance assumption of the ordinary logistic regression model. Hence, measurements are nested within an individual; we expect that measurements within the same individuals are more likely to be related to each other than the other individuals. To assess the nested effect, intra-class correlation coefficient was computed as; $\frac{\sigma_\mu^2}{\sigma_\mu^2 + \pi^2/3}$, where: the ordinary logit distribution has variance of $\pi^2/3$, $\sigma_\mu^2$ indicates the cluster variance [28]. The calculated ICC was 11.56% in the null model while $\sigma_\mu^2$ was 0.43. This implies that there is a need to take into account the between individual variability by using advanced models. Therefore, for the associated factors, we used the binary logistic mixed-effect regression model. Likelihood Ratio (LR) test and Deviance (-2LLR) was used for model comparison. Accordingly, a mixed effect logistic regression model (both fixed and random effect) was the best-fitted model since it had the lowest deviance value. Both bi-variables and multivariable binary logistic regression models were considered. Variables with a p-value < 0.2 in the bi-variable analysis were considered in the multivariable. Since the mixed effect model was estimating subject-specific estimates, we covert to population average by a conversion factor: $\frac{1}{\sqrt{(1+0.346\delta_b^2)}}\beta$ (where; $\beta$ = fixed effect and $\delta_b^2$ = random effect estimate) for the interpretation purpose. Finally, both crude and adjusted odds ratios with a 95% Confidence Interval (CI) of the selected model were reported. P-value $\leq$ 0.05 in the multivariable model were used to declare significant factors associated with poor treatment outcomes.

## Ethical consideration

The University of Gondar's institutional review board provided ethical approval. The Amhara public health institution sent a letter of support, and the medical directors of each hospital approved. The goal, objectives, and right to participate or not engage in the study were all explained to the participants. Participants' permission to withdraw from the study at any time and without explanation was clearly stated. Before data collection, each subject gave their written consent. Furthermore, rather than using personal identifiers, code numbers were utilized to ensure confidentiality.

## Result

### Background characteristics

A total of 836Diabetic and/ or Hypertensive patients were included in the study with a response rate of 99%. The man diagnosis was Diabetes in 410 (49%) patients. The median age of the study participants was 52 years with Inter Quartile Range (IQR) of 18 (43–61). Nearly two-third of the study participants, 543 (65%) were urban dwellers. Besides, the median duration on follow-up was 5 years with an IQR of 5 (3–8) years. A quarter of the patients, 209 (25%), were housewife followed by a government employee (22%) in their employment. Although more than half (54%) of patients were covered by health insurance, 351 (42%) participants paid their medical expenses from their pocket. Three hundred thirty-six (40%) of patients had one or more identified co-morbidities. While109 (13%) of patients have one or more chronic complications. Five hundred thirty-five (64%) of patients take multiple prescribed medications.

Related to the habit of the study participants, 84 (10%) of our patients have a history of hazardous alcohol use. Moreover, three hundred eighteen (38%) of the patients didn't meet the WHO recommendation on physical activity for health.

RelatedCOVID-19 pandemic, 334 (40%) of patients had at least one emergency visit during the pandemic period. One in four, (23.02%), of the participants, had COVID-19 like symptom(s) during the study period but more than half of the study participants, 453 (54.988%), perceived that they had COVID-19 infection. Of those who had symptoms consistent with COVID-19, only 9 were tested positive for COVID-19 infection. Hypertensive and/or Diabetic patients missed their medical appointment(s) during the COVID-19 pandemic. Eighty-four patients (10%) missed appointment before COVID- 19 but 205 (31%) of them were missed their appointment due to the pandemic. Nearly half of the patients, 117(45%), who missed their appointments ascribed the missed visit for fear of acquiring COVID-19 infection from Hospital (Table 1).

**Table 1. Background characteristics of Diabetic and/ or Hypertensive patients in Northwest Ethiopia, 2021 (n = 836).**

| Variable | Category | Frequency | Percent |
|---|---|---|---|
| Main diagnosis | Diabetes | 410 | 49.04 |
|  | Hypertension | 426 | 50.96 |
| Age (mean±IQR) | - | 52± 20.5 (43, 61.5) | |
| Residence | Urban | 546 | 65.31 |
|  | Rural | 290 | 34.69 |
| Employment | Housewife | 208 | 24.88 |
|  | Government employee | 183 | 21.89 |
|  | Private employee | 54 | 6.46 |
|  | Farmer | 156 | 18.66 |
|  | Merchant | 89 | 10.65 |
|  | Unemployed | 50 | 5.98 |
|  | Student | 36 | 4.31 |
|  | Other | 60 | 7.18 |
| Payment method | Health Insurance | 454 | 54.31 |
|  | Out of Pocket | 351 | 41.99 |
|  | Waived | 17 | 2.03 |
|  | Poverty card | 14 | 1.67 |
| Marital status | Single | 81 | 9.69 |
|  | Married | 605 | 72.37 |
|  | Divorced | 75 | 8.97 |
|  | Widowed | 75 | 8.97 |
| Duration of follow up (mean±IQR) |  | 55 ± (3,8) | |
| Regimen medication | Oral | 2,805 | 67.11 |
|  | Injectable | 875 | 20.93 |
|  | Both | 500 | 11.96 |
| Frequency of medication taken per day | Daily | 84 | 10.05 |
|  | Twice (BID) | 572 | 68.42 |
|  | Three times (TID) | 69 | 8.25 |
|  | Four times (QID) | 48 | 5.74 |
|  | Five and above | 63 | 7.54 |
| Kinds of medication in number (mean±IQR) |  | 2± 2(1,3) | |
| Hazardous alcohol use | No | 753 | 90.07 |
|  | Yes | 83 | 9.93 |
| Sedentary life style | No | 539 | 64.47 |
|  | Yes | 297 | 35.53 |

(*Continued*)

**Table 1.** (Continued)

| Variable | Category | Frequency | Percent |
|---|---|---|---|
| Presence of co-morbidity | No | 335 | 40.56 |
| | Yes | 491 | 59.44 |
| Presence of complication | No | 717 | 86.91 |
| | Yes | 108 | 13.09 |
| Presence of COVID-19 like symptoms | No | 642 | 76.98 |
| | Yes | 192 | 23.02 |
| Perception COVID-19 infection | Most likely | 29 | 3.52 |
| | Likely | 39 | 4.73 |
| | I can't decide | 49 | 5.95 |
| | Less likely | 254 | 30.83 |
| | Never | 453 | 54.98 |
| Emergency visit during the pandemic | No | 719 | 86.00 |
| | Yes | 117 | 14.00 |
| Missed appointment before COVID-19 pandemic | No | 745 | 90.08 |
| | Yes | 82 | 9.92 |
| Missed appointment during COVID-19 pandemic | No | 569 | 69.05 |
| | Yes | 255 | 30.95 |

## Magnitudes of poor treatment outcome

The magnitudes of poor treatment control before the COVID- 19 pandemic was 24.81% (21.95, 27.92) but following the COVID-19 pandemic, poor treatment outcome was 30.33% (27.01, 33.88) in the first three months, 35.66% (32.26, 39.20) in the second three months, 36.69% (33.40, 40.12) in the third three months and34.18% (3102, 37.49) the fourth three months following the first date COVID-19 case the detection of in Ethiopia (Fig 1).

## Factors associated with poor treatment control

After identifying variables significantly associated with poor treatment control in the bi-variable analysis at a p-value less than 0.2.Variables such as marital status, duration of follow up, regimen of medications, frequency of drug use, number of medications, hazardous alcohol use, sedentary lifestyle, presence of complication, and missed appointment during the pandemic were significantly associated with poor treatment outcome among diabetic and/or hypertensive patients during COVID-19 pandemic at a 5% level of significance.

   This study revealed that the odds of poor treatment control among married participants were 44% (AOR = 0.56, 95%CI; 0.41, 0.74) lower as compared with unmarried participants. The likelihood of having poor treatment control for patients who take injectable medication was 1.30 (AOR = 1.30, 95%CI; 1.02, 166), times higher as compared with those who take oral medication. Besides, the frequency of drug use per day was significantly associated with poor treatment control. As compared with patients taking drug five or more times per day, the chance of poor treatment control was 88% (AOR = 0.12, 95%CI;0.08, 0.20), 58% (AOR = 0.42, 95%CI;0.30. 0.59), and 69% (AOR = 0.31, 95%CI; 0.21, 0.47) lower for patients taking one, two, and three times per a day, respectively. As the number of medications taken increased, the likelihood of having poor treatment outcome was reduced by 21% (AOR = 0.79, 95%CI; 0.73, 0.87). Patients' lifestyles played a great role in affecting poor treatment outcome. As compared to their counterparts, the odds of having poor treatment outcome was 1.29 (AOR = 1.29, 95%CI; 1.02, 1.65) and 1.72 (AOR = 1.72, 95% CI;1.46, 2.02) higher for patients having

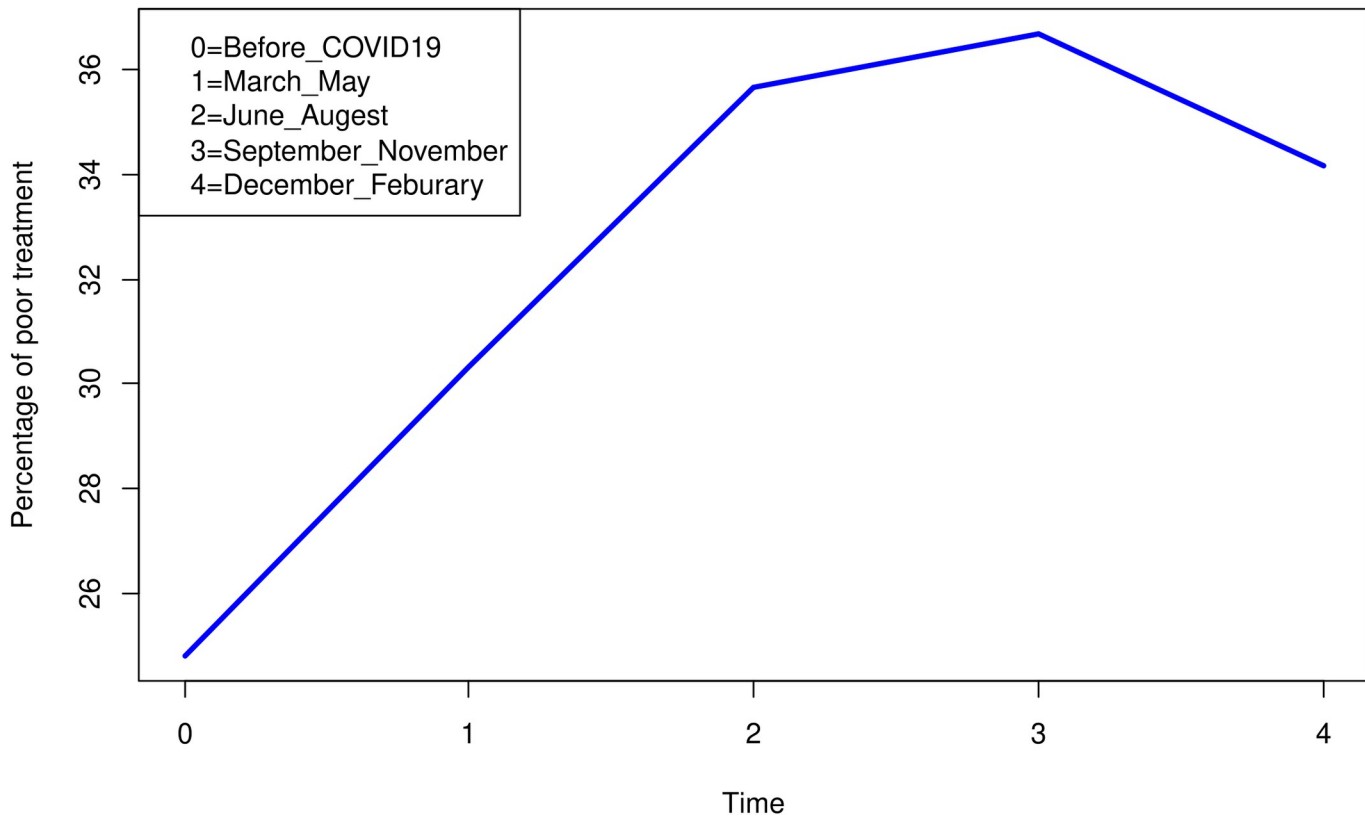

**Fig 1. The proportion of poor treatment outcomes among ambulatory Diabetic and/ or Hypertensive patients in Northwest Ethiopia, 2021 (n = 836).**

hazardous alcohol use and sedentary lifestyle, respectively. Missed appointment was an important and significant predictor for poor treatment control with the chance of having poor treatment outcome was two (AOR = 2.09, 95%CI; 1.79, 2.45) times higher for patients who have missed their appointment during the pandemic period as compared to their counterparts. The presence of disease-related complication(s) increases the likelihood of poor treatment control by 11% (AOR = 1.11, 95%CI; 0.93, 1.34) as compared to their counterparts (Table 2).

## Discussion

The impact of the COVID-19 pandemic on the healthcare system and patients' care has been sustained as the pandemic continues. Even though adherence to measures has been variable and inconsistent, Ethiopian governments have been forced to adopt legal measures to contain the spread of COVID-19 infection, including short-term complete lockdown, social distancing, prohibition of social gatherings, and school closure. The pandemic has had a significant impact on high-risk groups such as patients with chronic medical conditions such as hypertension and diabetes, either directly or indirectly [12–14].

The purpose of this study was to investigate the effect of COVID-19 on the magnitudes of poor treatment control among ambulatory Diabetic and/or Hypertensive patients and its associated factor using a generalized linear mixed model.

During the COVID-19 pandemic, the magnitudes of poor treatment increased significantly. This shift was most noticeable in the second and third three months following the first COVID-19 case detection in Ethiopia. This could be due to disruptions in regular care caused by restrictions on essential health service visits, which forced them to stay at home, as well as

**Table 2. Multivariable binary logistic regression mixed model for associated factors of Poor treatment control among ambulatory Diabetic and/or Hypertensive patients.**

| Variable | Category | COR (95%CI) | AOR (95%CI) |
|---|---|---|---|
| Age in years | | 0.992 (0.987, 0.997) | 0.997 (0.991, 1.01) |
| Residence | Urban | 1 | 1 |
| | Rural | 1.27 (1.10, 1.45) | 1.01 (0.86, 1.18) |
| Marital status | Single | 1.51(1.21, 1.870 | 1.19 (0.09, 1.58) |
| | Unmarried* | 1 | 1 |
| | married | 0.56 (0.48, 0.81) | 0.56 (0.42, 0.75)* |
| Duration of follow up in years | | 1.07 (1.06, 1.09) | 1.06 (1.05, 1.08)* |
| Types of medications | Oral | 1 | 1 |
| | Injectable | 1.92 (1.63, 2.26) | 1.30 (1.02, 166)* |
| | Both | 1.26 (1.02 1.55) | 1.27 (0.99, 1.62) |
| Frequency of drug use | Daily | 0.20 (0.13, 0.29) | 0.12 (0.08, 0.20)* |
| | BID | 0.86 (0.67, 1.11) | 0.42 (0.30, 0.59)* |
| | TID | 0.42 (0.29, 0.61) | 0.31 (0.21, 0.47)* |
| | QID | 0.86 (0.61, 1.25) | 0.71 (0.47, 1.07) |
| | > = 5 times | 1 | 1 |
| Number of medications | | 0.86 (0.81, 0.91) | 0.79 (0.73, 0.87)* |
| Hazardous alcohol use | no | 1 | 1 |
| | Yes | 1.28 (1.03, 1.59) | 1.29 (1.02, 1.65)* |
| Sedentary life style | No | 1 | 1 |
| | Yes | 1.82 (1.59, 2.09) | 1.72 (1.46, 2.02)* |
| Missed appointment during the pandemic | No | 1 | 1 |
| | Yes | 2.36 (2.04, 2.73) | 2.09 (1.79, 2.45)* |
| Presence of co-morbidity | No | 1 | 1 |
| | Yes | 1.39 (1.21, 1.61) | 1.11 (0.93, 1.34) |
| Presence of co-morbidity | No | 1 | 1 |
| | Yes | 1.28 (1.08, 1.32) | 1.27 (1.01, 1.59)* |
| Group level variance | | 0.053 | |
| PCV (%) | | 87.67 | |
| Deviance | | 4236.74 | |

Unmarried*—single, widowed, and divorced

*P-value <0.05

limiting physical contact with health professionals. When compared to the pre-pandemic period, the magnitudes of individuals who had missed their appointments increased more than twice during the pandemic period. As a result, patients who do not have strict follow-up and miss their appointments are more likely to have poor treatment control.

In multivariable binary logistic mixed model, several factors such as marital status, duration of follow up, presence of complication, types of medication, frequency of medication used per day, numbers of medication, behavioural factors such as hazardous alcohol use and sedentary lifestyle, and missing of appointment were significantly associated with poor treatment control among ambulatory Diabetic and/or Hypertensive patients.

According to the findings of this study, married participants had a lower chance of having poor treatment than unmarried participants. This finding, however, was consistent with previous studies conducted in Pakistan and China, which found that being married reduces the likelihood of poor treatment control [15, 16]. The possible reason for this finding could be that

married individuals might get support from their partners which positively affects the adherence to control measures for their underlying conditions [17].

In agreement with the previous studies [18, 19], this study also strikes that duration follow-up was the important and significant factor for poor treatment control among ambulatory Diabetic and/or Hypertensive patients. This could be justified by as the duration of follow-up increases; the chance to develop disease-related complications will be higher which results in poor disease control. Besides, having a longer duration of follow-up might compromise the patient's beliefs about the effectiveness of medication and control measures [20]. Therefore, it is better to screen for disease-related complications to achieve good disease control.

In terms of medications, those who take injectable medications had a higher risk of poor treatment control than those who took PO medications. However, the types of medications and the frequency with which they were taken per day were negatively associated with poor treatment control. Patients taking one (daily), two (BID), three (TID), and four (QID) times per day had a lower risk of poor treatment control than those taking five or more times per day. Furthermore, a patient who took more types of medication had a lower chance of having poor treatment control. Previous studies supported this evidence [21, 22]. This could be explained by that dosage and regimen for administration of medication had a paramount effect on medication adherence which is vital for controlling disease progressions and preventing complications [23], there is also a fact that patients taking injections may find it difficult to adjust treatment without the immediate support of health care providers [5, 11, 24]. Besides, taking multiple medications had a synergistic effect for treating the disease and preventing its complications [25].

Hazardous alcohol use and sedentary behaviour were also the significant factors of poor treatment control among ambulatory Diabetic and/or Hypertensive patients. It is in agreement with previous studies [26–28]. Because there is no safe alcohol range for chronic disease patients, and physical exercise is an important component of lifestyle modification, lifestyle modification plays a significant role in chronic disease management, with the ultimate goal of preventing disease progression and related complications in instances where a complete cure cannot be achieved [29, 30].

Missed appointment during the pandemic period was significantly and strongly associated with poor treatment control with the likelihood of having poor treatment control for patients who have missed their appointments were two times higher as compared to their counterparts. This finding was supported by previous studies [31, 32]. This could be justified by the fact that missed appointment is one dimension of adherence where strict adherence to medication and appointment is required for chronic disease management [33].

The current study used internationally and/or locally validated tools for measuring physical activity and hazardous alcohol use, and data were collected by trained and experienced nurses and medical doctors under close and supportive supervision. The respondents were also informed about the importance of the study and the confidentiality of personal data to gain the trust of respondents and minimize the non response rate. But this study was not free of limitations. The study includes two medical conditions, thus the factors for poor treatment control might be different for each disease entity attention should be given while interpreting the findings of the study. Since the study was facility-based there might be a risk of social desirability bias. Moreover, there might be a risk of misclassification bias because the outcome variables were ascertained by the physician assessment.

## Conclusion

COVID-19 pandemic has significantly affected the treatment control of ambulatory Diabetic and/ or Hypertensive patients. Being married, the frequency and kinds of drugs taken per day

were negatively associated with treatment control. Whereas hazardous alcohol use, sedentary lifestyle, longer duration of follow up, having a disease-related complication, patients taking injectable medication, per day, and missed appointments during the pandemic of COVID -19 were positively associated with poor treatment control of ambulatory Diabetic and/ or Hypertensive patients. Therefore, it is better to consider the risk factors of poor treatment control while designing and implementing policies and strategies for chronic disease control.

## Supporting information

**S1 File.**
(DTA)

## Acknowledgments

The authors would like to thank the University of Gondar, Amhara Health Bureau, and Amhara Public Health Institute for the technical support and facilitation they provide during the study. We also thank the study participants for providing the information during the interview.

## Author Contributions

**Conceptualization:** Tadesse Awoke Ayele, Habtewold Shibru, Malede Mequanent Sisay, Tesfahun Melese, Melkitu Fentie, Tariku Belachew, Kegnie Shitu, Tesfa Sewunet Alamneh.

**Data curation:** Tadesse Awoke Ayele, Malede Mequanent Sisay, Kegnie Shitu, Tesfa Sewunet Alamneh.

**Formal analysis:** Tadesse Awoke Ayele, Habtewold Shibru, Malede Mequanent Sisay, Melkitu Fentie, Tesfa Sewunet Alamneh.

**Funding acquisition:** Tadesse Awoke Ayele.

**Investigation:** Tadesse Awoke Ayele, Habtewold Shibru, Malede Mequanent Sisay, Tesfahun Melese, Melkitu Fentie, Telake Azale, Tariku Belachew, Kegnie Shitu, Tesfa Sewunet Alamneh.

**Methodology:** Tadesse Awoke Ayele, Habtewold Shibru, Malede Mequanent Sisay, Tesfahun Melese, Melkitu Fentie, Telake Azale, Tariku Belachew, Kegnie Shitu, Tesfa Sewunet Alamneh.

**Project administration:** Tadesse Awoke Ayele.

**Resources:** Tadesse Awoke Ayele, Habtewold Shibru, Tesfahun Melese, Tariku Belachew.

**Software:** Tadesse Awoke Ayele, Malede Mequanent Sisay, Tesfahun Melese, Melkitu Fentie, Kegnie Shitu, Tesfa Sewunet Alamneh.

**Supervision:** Tadesse Awoke Ayele, Melkitu Fentie, Tariku Belachew, Kegnie Shitu, Tesfa Sewunet Alamneh.

**Validation:** Habtewold Shibru, Telake Azale, Kegnie Shitu.

**Visualization:** Tadesse Awoke Ayele, Habtewold Shibru, Malede Mequanent Sisay, Kegnie Shitu, Tesfa Sewunet Alamneh.

**Writing – original draft:** Tadesse Awoke Ayele, Habtewold Shibru, Malede Mequanent Sisay, Tesfahun Melese, Melkitu Fentie, Kegnie Shitu, Tesfa Sewunet Alamneh.

**Writing – review & editing:** Tadesse Awoke Ayele, Habtewold Shibru, Malede Mequanent Sisay, Tesfahun Melese, Melkitu Fentie, Telake Azale, Tariku Belachew, Kegnie Shitu, Tesfa Sewunet Alamneh.

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
