## [Decision Letter · Decision Letter 0]

9 Feb 2022

PONE-D-21-37762The effect of COVID-19 on poor treatment control among ambulatory Hypertensive and/or Diabetic patients in Northwest EthiopiaPLOS ONE

Dear Dr. Alamneh,

Thank you for submitting your manuscript to PLOS ONE. After careful consideration, we feel that it has merit but does not fully meet PLOS ONE’s publication criteria as it currently stands. Therefore, we invite you to submit a revised version of the manuscript that addresses the points raised during the review process.

We look forward to receiving your revised manuscript.

Kind regards,

Frank T. Spradley

Academic Editor

PLOS ONE

“The authors are grateful to the federal ministry of health (MoH) for sponsoring this research.”

Reviewers' comments:

Reviewer's Responses to Questions

**Comments to the Author**

1. Is the manuscript technically sound, and do the data support the conclusions?

Reviewer #1: Partly

2. Has the statistical analysis been performed appropriately and rigorously? 

Reviewer #1: I Don't Know

3. Have the authors made all data underlying the findings in their manuscript fully available?

Reviewer #1: Yes

4. Is the manuscript presented in an intelligible fashion and written in standard English?

Reviewer #1: No

5. Review Comments to the Author

Reviewer #1: 

1. A directional hypothesis needs to be stated that indicates what the authors expected to find. For instance, was it hypothesized that COVID-19 would worse outcomes?

2. It is not understood what is meant by “treatment control of ambulatory hypertensive and diabetic patients”. Does this mean access to doctors/medications? What types of medications are these patients on?

3. What is novel about this study? This should be highlighted in the abstract and introduction.

4. The introduction is too long and should be more succinct.

5. Inclusion of a biostatistician would greatly increase the rigor of this study.

6. The data need to be presented as figures to stress the point. Currently, Figure 1 does not express any statistics. And it would be helpful to have time on the x-axis.

7. It would be helpful to have some type of control group. Are more wealthy patients at an advantage?

8. It seems as if data could be analyzed throughout 2021.

9. Are there any data on pregnancy outcomes?

10. Impact of each of the variants?

11. Did any of these patients die of COVID?

12. This manuscript must be copyedited.

6. PLOS authors have the option to publish the peer review history of their article (what does this mean?). If published, this will include your full peer review and any attached files.

Reviewer #1: No

---

## [Author Response · Author response to Decision Letter 0]

1 Mar 2022

February 2022

Rebuttal letter

Manuscript ID: PONE-D-21-39918

Title: the effect of COVID-19 on poor treatment control among ambulatory Hypertensive and/or Diabetic patients in Northwest Ethiopia

Tadesse Awoke Ayele, HabtewoldShibru, Malede Mequanent, TesfahunMelese, MelkituFentie, TelakeAzale, Tariku Belachew, KegnieShitu, and TesfaSewunetAlamneh*

PLOS ONE

Dear Editor and reviewer, 

We would like to thank for your consideration and suggestion for the betterment of our manuscript and make it more informative. We tried to amend the format of the manuscript according to the journal guidelines and address the questions raised by reviewer on the manuscript. The authors revised the overall manuscript regarding to language usage and grammar errors. In addition, we also consult language experts in our university and amendments were done based on their comments. Our point-by-point responses for each comment and questions are described in detail on the following pages. Further, the details of changes were shown by track changes in the supplementary document attached. 

Editor’s comment

1. Please ensure that your manuscript meets PLOS ONE's style requirements, including those for file naming. The PLOS ONE style templates can be found athttps://journals.plos.org/plosone/s/file?id=wjVg/PLOSOne_formatting_sample_main_body.pdf andhttps://journals.plos.org/plosone/s/file?id=ba62/PLOSOne_formatting_sample_title_authors_affiliations.pdf

Authors’ response: Thank you dear editor for your concern. We tried to adjust the format according to the journal requirements

Authors’ response: Thank you dear editor for your concern. We corrected the grant numbers in the funding information section. 

3. Thank you for stating the following in the Acknowledgments Section of your manuscript: “The authors are grateful to the federal ministry of health (MoH) for sponsoring this research.” We note that you have provided additional information within the Acknowledgements Section that is not currently declared in your Funding Statement. Please note that funding information should not appear in the Acknowledgments section or other areas of your manuscript. We will only publish funding information present in the Funding Statement section of the online submission form. Please remove any funding-related text from the manuscript and let us know how you would like to update your Funding Statement. Currently, your Funding Statement reads as follows: “The funders had no role in study design, data collection and analysis, decision to publish, or preparation of the manuscript.” Please include your amended statements within your cover letter; we will change the online submission form on your behalf

Authors’ response: Thank you dear editor for your concern. We have removed it from the main document and included the details of funding information on the online submission system.

4. In your Data Availability statement, you have not specified where the minimal data set underlying the results described in your manuscript can be found. PLOS defines a study's minimal data set as the underlying data used to reach the conclusions drawn in the manuscript and any additional data required to replicate the reported study findings in their entirety. All PLOS journals require that the minimal data set be made fully available. For more information about our data policy, please see http://journals.plos.org/plosone/s/data-availability.Upon re-submitting your revised manuscript, please upload your study’s minimal underlying data set as either Supporting Information files or to a stable, public repository and include the relevant URLs, DOIs, or accession numbers within your revised cover letter. For a list of acceptable repositories, please see http://journals.plos.org/plosone/s/data-availability#loc-recommended-repositories. Any potentially identifying patient information must be fully anonymzed. Important: If there are ethical or legal restrictions to sharing your data publicly, please explain these restrictions in detail. Please see our guidelines for more information on what we consider unacceptable restrictions to publicly sharing data: http://journals.plos.org/plosone/s/data-availability#loc-unacceptable-data-access-restrictions. Note that it is not acceptable for the authors to be the sole named individuals responsible for ensuring data access. We will update your Data Availability statement to reflect the information you provide in your cover letter.

Authors’ response: Thank you dear editor for your concern. We have presented the appropriate data availability statement on the online submission

Authors’ response: Thank you dear editor for your concern. We have deleted it according to your recommendation. 

To reviewer 1

1. Directional hypothesis needs to be stated that indicates what the authors expected to find. For instance, was it hypothesized that COVID-19 would worse outcomes?

Authors’ response: Thank you dear reviewer for your concern. As you know patients with chronic disease has higher risk of developing severe form of COVID-19 and disease related deaths. In addition, there were different measures that are taken to prevent and control the spread of COVID-19. Therefore, COVID-19 would worse outcomes in patients either direct or indirect forms. As per your recommendation we have putted this hypothesis in the background section.

2. It is not understood what is meant by “treatment control of ambulatory hypertensive and diabetic patients”. Does this mean access to doctors/medications? What types of medications are these patients on?

Authors’ response: Thank you dear reviewer for your concern. Patients with chronic disease need a strict follow-up to prevent disease progression and related complications in out-patient. In some cases they might need admissions and they should receive inpatient care. Ambulatory in this case was used to indicate patients who take out-patient services.

3. What is novel about this study? This should be highlighted in the abstract and introduction.

Authors’ response: Thank you dear reviewer for your concern. We have putted it in the background section of abstract.

4. The introduction is too long and should be more succinct.

Authors’ response: Thank you dear reviewer for your concern. We revised the introduction based on your comments and tried to make it as short as possible while it is informative.

5. Inclusion of a biostatistician would greatly increase the rigor of this study.

Authors’ response: Thank you dear reviewer for your concern. At the beginning it includes biostatistician (i.e. Tadesse Awoke Ayele; professor of Biostatistics, Malede Mekuanit Sisay; assistant professor of biostatistics and currently PhD candidate at Utrecht University, and Tesfa Sewunet Alamneh; lecturer of biostatistics)

6. The data need to be presented as figures to stress the point. Currently, Figure 1 does not express any statistics. And it would be helpful to have time on the x-axis.

Authors’ response: Thank you dear reviewer for your concern. We have changed the figure which include time frame on the x-axis.

7. It would be helpful to have some type of control group. Are more wealthy patients at an advantage?

Authors’ response: Thank you dear reviewer for your concern. Yes indeed, considering control group could help to see the effect clearly, however it was difficult to use control groups while conducting the study. Regarding wealth status, we included the variable income as proxy indicator of wealth. However, this variable differs intrinsically between urban and rural residents. The income of rural residents should be measured using wealth index variables instead of income as they do not have monthly salary unlike employed urban residents. Because of this we dropped the variable income in the analysis.

8. It seems as if data could be analyzed throughout 2021.

Authors’ response: Thank you dear reviewer for your concern. We presented the data collection period which was from January to March 2021 but not entirely 2021

9. Are there any data on pregnancy outcomes?

Authors’ response: Thank you dear reviewer for your concern. We didn’t collect pregnancy outcomes related data because we collect data from chronic care clinic and we didn’t have access for pregnancy outcome related that are found in obstetrics clinic.

10. Impact of each of the variants?

Authors’ response: Thank you dear reviewer for your concern. As we said earlier the data was collected from January to March 2021. During this period there was one variant of COVID-19 in our country. Due to this, we were not able to assess the impact of each variant. 

11. Did any of these patients die of COVID?

Authors’ response: Thank you dear reviewer for your concern. One of the data collection techniques for conducting this study was interviewing the patients with chronic disease who were coming for follow-up at chronic clinic. To be included for our study, the patient needs to visit the chronic follow up during data collection period so the patient should in life to be one of the participants.

12. This manuscript must be copyedited.

Authors’ response: Thank you dear reviewer for your concern. The manuscript went through copyediting by language editors and the authors.

---

## [Decision Letter · Decision Letter 1]

21 Mar 2022

The effect of COVID-19 on poor treatment control among ambulatory Hypertensive and/or Diabetic patients in Northwest Ethiopia

PONE-D-21-37762R1

Dear Dr. Alamneh,

We’re pleased to inform you that your manuscript has been judged scientifically suitable for publication and will be formally accepted for publication once it meets all outstanding technical requirements.

Kind regards,

Frank T. Spradley

Academic Editor

PLOS ONE

Reviewers' comments:

Reviewer's Responses to Questions

**Comments to the Author**

1. If the authors have adequately addressed your comments raised in a previous round of review and you feel that this manuscript is now acceptable for publication, you may indicate that here to bypass the “Comments to the Author” section, enter your conflict of interest statement in the “Confidential to Editor” section, and submit your "Accept" recommendation.

Reviewer #1: All comments have been addressed

2. Is the manuscript technically sound, and do the data support the conclusions?

Reviewer #1: Yes

3. Has the statistical analysis been performed appropriately and rigorously? 

Reviewer #1: Yes

4. Have the authors made all data underlying the findings in their manuscript fully available?

Reviewer #1: Yes

5. Is the manuscript presented in an intelligible fashion and written in standard English?

Reviewer #1: Yes

6. Review Comments to the Author

Reviewer #1: The authors have adequately addressed my previous comments and concerns. This is now suitable for publication.

7. PLOS authors have the option to publish the peer review history of their article (what does this mean?). If published, this will include your full peer review and any attached files.

Reviewer #1: No

---

## [Editor Report · Acceptance letter]

25 Mar 2022

PONE-D-21-37762R1 

The effect of COVID-19 on poor treatment control among ambulatory Hypertensive and/or Diabetic patients in Northwest Ethiopia 

Dear Dr. Alamneh:

I'm pleased to inform you that your manuscript has been deemed suitable for publication in PLOS ONE. Congratulations! Your manuscript is now with our production department. 

Kind regards, 

on behalf of

Dr. Frank T. Spradley 

Academic Editor

PLOS ONE